# Digital Eye Strain among Peruvian Nursing Students: Prevalence and Associated Factors

**DOI:** 10.3390/ijerph20065067

**Published:** 2023-03-13

**Authors:** Sonia Celedonia Huyhua-Gutierrez, Jhon Alex Zeladita-Huaman, Rosa Jeuna Díaz-Manchay, Albila Beatriz Dominguez-Palacios, Roberto Zegarra-Chapoñan, María Angélica Rivas-Souza, Sonia Tejada-Muñoz

**Affiliations:** 1Academic Department of Public Health, Institute of Tropical Diseases, Universidad Nacional Toribio Rodríguez de Mendoza de Amazonas, Amazonas, Chachapoyas 01001, Peru; 2Academic Department of Nursing, Universidad Nacional Mayor de San Marcos, Lima 15001, Peru; 3Faculty of Medicine, Universidad Católica Santo Toribio de Mogrovejo, Lambayeque 14001, Peru; 4Faculty of Health Science, Universidad Nacional de Cajamarca, Cajamarca 06001, Peru; 5Faculty of Health Science, Universidad María Auxiliadora, Lima 15408, Peru; 6Committee of Eye Health and Prevention of Blindness, Ministry of Health, Lima 15072, Peru

**Keywords:** nursing students, digital eye strain, computers, COVID-19

## Abstract

There has been a high prevalence of digital eye strain (DES) among students who have received distance-learning lessons due to COVID-19. However, in low- and middle-income countries, there are few studies that have analyzed its associated factors. This study aimed to determine the prevalence of DES and its associated factors among nursing students during COVID-19 distance learning. This was a cross-sectional analytical study conducted between May and June 2021 in six Peruvian universities. The sample comprised 796 nursing students. DES was measured using the Computer Vision Syndrome Questionnaire (CVS-Q). A bivariate logistic regression analysis was performed. DES was found in 87.6% of nursing students. Sitting upright (OR, 0.47; 95% IC, 0.30–0.74), using electronic devices for more than four hours a day (OR, 1.73; 95% IC, 1.02–2.86), not following the 20-20-20 rule (OR, 2.60; 95% IC, 1.25–5.20), having the screen brightness very high (OR, 3.36; 95% IC, 1.23–11.8), and not wearing glasses (OR, 0.59; 95% IC, 0.37–0.93) are factors associated with DES. The prevalence of DES among nursing students is high. Improving the ergonomics of study environments, reducing the time of exposure to electronic devices, adjusting the screen brightness, and taking eye-care measures are key to controlling computer vision syndrome in virtual learning.

## 1. Introduction

To avoid the spread of COVID-19, most countries imposed distancing measures such as travel restrictions and border closures [1]. In the educational sphere, they transitioned from in-person learning to distance learning [2]. Particularly in health education, institutions implemented electronic pedagogical innovations and prioritized simulation-based teaching to ensure the well-being of students during virtual learning [3]. Although virtual learning offers diverse benefits, such as access to online commercial platforms and teleconference systems, it demands increased exposure to electronic screens [4] and, therefore, the extended use of information and communication technology (ICT) tools, including cell phones, tablets, and computers. This undermines students’ eye health [5,6,7,8].

One of these under-researched eye-health problems, which relates to the impact of COVID-19 on education, is digital eye strain (DES) [9]. Its symptoms include blurred vision, difficulty focusing, irritation, dry eyes, visual fatigue, headaches, and increased sensitivity to light. These symptoms hinder students’ learning process, as they could experience progressive and irreversible eye injuries [4,7,10,11,12]. In addition, the prevalence of DES among university students that received virtual lessons was higher than in other population groups [8].

Studies conducted in Europe and Asia reported that between 70% and 90% of university students—specifically health-sciences students—suffer from DES [9,11,13,14]. However, few studies have been carried out in this regard in low- and middle-income countries [15]. Furthermore, although the literature agrees on the relationship between the time spent using electronic devices and the prevalence of DES, there are different results concerning the association of DES with sociodemographic characteristics; these include age and sex, as well as its association with conditions that are related to the use of electronic devices, such as brightness intensity, the distance of the screen, the use of monitor filters, and breaks taken [7,9,11,16,17,18,19]. This could be the result of different reporting methods. Hence, in this study, we aimed to determine DES prevalence and its associated factors among nursing students during COVID-19 distance learning.

## 2. Materials and Methods

### 2.1. Design, Duration, and Approval

This was a quantitative, relational, analytical, and cross-sectional study [20] conducted between May and July 2021. It was approved by the Ethics Committee of Universidad Nacional Toribio Rodríguez de Mendoza de Amazonas through Letter No. 001-2021. The first part of the questionnaire consisted of the informed consent, in which students voluntarily decided to participate in the study.

### 2.2. Population, Sample and Selection Criteria

The population comprised 1945 nursing students from six universities (four public and two private) located in the three natural regions (the coast, highlands, and rainforest) of Peru. The students were enrolled in the first through the tenth academic cycle during the 2021 academic year, were 18 years of age or older, and had an electronic device with an internet connection. The estimated sample was 796 participants, with an estimated power of at least 0.80, a Type I Error probability set to 0.05, and a small effect size. The sample was selected by snowball sampling [20].

### 2.3. Measurement Tools

To identify DES, we administered the Computer Vision Syndrome Questionnaire (CVS-Q), which is used in studies with university students [6]. This Likert scale was adapted for and validated by the Peruvian population, showing a good internal consistency (Cronbach’s alpha was 0.87) [15]. It measured the frequency and intensity of nine DES symptoms. The response options to determine the frequency were never (not once) = 0; sometimes (once a week) = 1; and always (two or three times a week) = 2. We included dichotomous questions to determine the intensity: moderate = 1 and intense = 2. We classified the total score on two levels: with DES (<6) and without DES (≥6) [9,21].

Moreover, the study considered the conditions of using electronic devices, such as the type of device most used for class; the number of hours per day spent using the device; the distance between the device and the eyes; the posture adopted when using the device; screen brightness; the use of monitor filters if computers, laptops, or tablets were used; frequency of use; and duration of breaks. Likewise, students were asked if they knew and practiced the 20-20-20 rule and if they wore glasses. The technique used was online surveying, and the instrument was a survey prepared on Google Forms. We sent the link to the questionnaire via e-mail, which students answered within approximately 15–20 min. Directors at the universities who were involved in the study approved the data collection.

### 2.4. Statistical Analysis

First, we conducted a descriptive analysis of the study variables. Subsequently, we performed a logistic regression analysis to identify factors associated with the presence or absence of DES among nursing students. We chose the associated factors through a stepwise algorithm that determined, based on changes in the Akaike information criterion, the variables that had a better predictive capacity on the response variable. We ran the outlier detection technique and tested the assumptions in the resulting model, through which we observed that no case was atypical and confirmed all the assumptions of the logistic regression. For the interpretation of the logistic regression parameters, it was necessary to exponentiate the coefficients and read them as odds. We performed all the analyses on the software R v4.0.1. The level of significance established in the study was 0.05.

## 3. Results

### 3.1. Participant Characteristics

The participants were 796 nursing students from six Peruvian universities, of which 641 (80.53%) were women and 155 (19.47%) were men, 540 (67.84%) were 20–29 years old, and 358 (44.097%) were studying and working at the same time. In addition, 132 (16.42%) participants reported having children.

### 3.2. Conditions of Using Electronic Devices

Regarding the type of electronic device most frequently used to receive their classes during COVID-19 distance learning, 627 (78.77%) participants used a computer, 159 (19.97%) used a smartphone, and only 10 (1.26%) used a tablet. Regarding the daily hours of device usage, 671 (84.3%) participants spent more than 4 h, while the rest of the participants used their devices for less than 4 h. In addition, 440 (55.40%) participants used their devices at a distance from 30 to 50 cm; however, 315 (39.57%) used their devices at a distance less than 30 cm, while 40 (5.03%) participants used theirs at more than 50 cm. Regarding the posture that they usually adopted when using their devices, 514 (64.57%) students remained seated and bent over, 262 (32.91%) sat upright, and only 29 (2.51%) reported laying down. Likewise, regarding screen brightness, 520 (65.33%) participants used their device in dull mode 177 (22.24%); however, 99 (12.44%) participants used theirs in very bright mode. Regarding the use of glasses when using devices, only 385 (48.37%) reported using them. On the other hand, regarding the frequency of breaks, 269 (33.79%) participants reported that they rested at an interval greater than 2 h; 225 (28.27%) every 30 min; 240 (30.15%) every two hours; and 122 (15.33%) every hour. Regarding the use of monitor filters, 594 (74.50%) students did not use one, while 121 (15.20%) sometimes used one, and only 82 (10.30%) used one frequently. Regarding the knowledge and practice of the 20-20-20 rule, only 105 (13.19%) reported knowing this preventive measure, and only 50 (6.28%) students used it.

### 3.3. Prevalence of Digital Eye Strain

It was found that 697 (87.56%) nursing students presented with DES; meanwhile, 99 (12.44%) participants did not suffer from this ocular health problem.

### 3.4. Bivariate Analysis

Table 1 presents the bivariate analysis. Specifically, it can be observed that there are statistically significant differences in the presence and absence of DES according to the number of hours per day that students use electronic devices (*p* = 0.001), the posture they adopt when using the devices (*p* < 0.001), the brightness intensity they choose (*p* = 0.011), their use of glasses (*p* = 0.014), and their awareness (*p* = 0.041) and observance of the 20-20-20 rule (*p* = 0.001). As for the other characteristics, we observed no statistically significant differences.

### 3.5. Factors Associated with Digital Eye Strain

Table 2 summarizes the logistic regression model that results from the stepwise variable selection for associated factors of DES. For this model, the algorithm selected six predictor variables: posture when using the devices, the daily hours spent using devices, observance of the 20-20-20 rule, brightness, the use of glasses, and monitor filters. We observed that the students who used devices when they were sitting upright had a risk of DES (presence of DES) 0.47 times lower than those who used them when they were sitting hunched forward (b = −0.76; OR = 0.47 [0.30–0.74]; *p* < 0.01). Furthermore, the students who used the devices for more than four hours had a risk of DES 1.73 times higher than those who used them for less than four hours (b = 0.55; OR = 1.73 [1.02–2.86]; *p* = 0.04). Likewise, the students who did not follow the 20-20-20 rule had a risk of DES 2.60 times higher than those who observed the rule (b = 0.95; OR = 2.60 [1.25–5.20]; *p* = 0.01). The students who kept the devices in a very bright mode had a risk of DES 3.36 times higher than those who regulated their devices in dull mode (b = 1.21; OR = 3.36 [1.23–11.8]; *p* = 0.03). For their part, the students who did not wear glasses had a risk of DES 0.59 times lower compared to those who wore glasses (b = −0.52; OR = 0.59 [0.37–0.93]; *p* = 0.03). Finally, there was no statistically significant difference between those who sometimes used filters and those who never used them.

## 4. Discussion

The main finding of this study conducted with nursing students from six Peruvian universities who received distance-learning lessons due to the COVID-19 pandemic was that increased hours on electronic devices and intense screen brightness raised the risk of DES. On the other hand, the students who wore glasses followed the 20-20-20 rule, and adopted an upright sitting posture when using electronic devices had a lower risk of DES. This finding proves that eye health was negatively impacted by increased screen time as a result of the changes in learning modality that were reported among university distance-learning students, as well as the mass use of ICT tools, thanks to their educational benefits. However, there are ergonomic factors and preventive practices that could reduce this problem.

There is no consensus among researchers concerning the number of symptoms (ocular and non-ocular) that are measured and the method that is considered to determine DES prevalence (through a score or with at least one symptom). Consequently, the prevalence of DES indicated in this paper is similar to studies conducted among university students and professionals during the COVID-19 pandemic that included 15–16 symptoms and considered a score to determine DES prevalence [9,17]. However, the prevalence of DES that was revealed by this study is lower than that reported in studies that considered the presence of DES with at least one symptom [7,22]. In turn, it is higher compared to studies that considered between six and nine symptoms [11,13,17,23,24]. Nevertheless, the high DES prevalence among Peruvian nursing students reported here is consistent with studies carried out among university students in other countries during the COVID-19 pandemic [9,17] and professionals who worked in front of a computer [16,22].

One of the aspects that could explain this issue is the change in learning modality from in-person to distance learning: a context that boosts the use of electronic devices. This could eventually lead to increased refractive errors and ocular symptoms such as DES [8]. Consequently, it is paramount that higher education institutions take preventive measures in the use of digital screens; promote correct postures, eye care, and proper blinking; and allow ophthalmologists to conduct assessments for the correction of refractive errors and the treatment of eye diseases that could otherwise get worse. Likewise, researchers should further study the possible compromise of eye health in the different age groups exposed to DES.

As expected, finding that nursing students who used electronic devices for more than four hours have a higher risk of DES than those who used them for fewer hours confirms the scientific consensus that the hours spent using electronic devices are an associated factor for the prevalence of DES. This has been determined by studies that employed robust statistical analyses (multivariate analyses) [9,16,18,25]. However, studies that considered a wider range of hours for electronic device usage disagreed with this finding. In this regard, a bivariate-analysis study (using the Mann–Whitney U test) conducted before the COVID-19 pandemic with Saudi Arabian health-sciences students showed that using electronic devices for more than six hours was not associated with increased symptoms of DES [13]. Another study conducted in Indonesia highlighted, in the bivariate analysis, that using a visual display terminal for more than six hours was an associated factor, while in the multivariate analysis, this factor was not a predictor of DES prevalence [26].

Another associated factor of DES reported in this study is screen brightness. Specifically, students who use screens in the very bright mode have a risk of DES that is three times higher than those who use it in dull mode. However, there is no significant difference in the risk of DES among those who leave the brightness mode on compared to those who keep the dull mode on. This risk factor was also revealed by studies conducted with Saudi Arabian medical students [7], Thai university students [17], Indian students and teachers [8], and Ethiopian workers at government agencies [21]. The latter study only pointed out statistical differences in the bivariate analysis and not in the multivariate analysis.

Some studies that only performed bivariate analyses and considered a smaller sampling size suggested no association between brightness and DES prevalence. This was the case for two studies: one conducted with medicine and business students from Saudi Arabia, which excluded those with a history of eye issues, and considered four levels of brightness intensity (very bright, bright, dull, and very dull) [11]. The other study was conducted with Indonesian nursing students and considered two levels of intensity (dark and light) [12]. In this sense, studies that measure this association should establish uniform criteria, such as dividing the device brightness bar into thirds and standardizing the brightness intensity.

Furthermore, there are still contradictions in the association of screen brightness adjustment with DES. While a study conducted with information technology professionals in Egypt reported that adjusting screen brightness was a DES-preventive factor [18], another study that was carried out with employees at an Ethiopian university revealed that it was not a risk factor [19]. In addition, the analysis of the association of screen brightness with DES prevalence should consider lighting conditions (window glare, ceiling lighting, wall or ceiling reflection, and reflection on the computer screen), as these could also cause a huge difference in brightness in the visual field and lead to eye discomfort.

Based on the findings of this study and the existing scientific evidence from studies conducted with university students [7,8,17], it is recommended that people who use electronic devices keep the screen in dull mode, adjust screen/environment brightness to reduce reflection or glare, and have optical compensation with appropriate refractive error correction [18,24,27]. Moreover, if ocular or non-ocular symptoms of DES remain, people should request a specialized assessment.

Regarding DES preventive factors, this paper has shown that students who use devices sitting upright have a lower risk of DES than those who use them sitting while bending their backs. This is consistent with a study undertaken with employees at a university in Ethiopia [19]. However, we found no differences in the risk of DES between people who use devices lying down and those who use them sitting bent over, which is consistent with previous studies that measured this factor among Saudi Arabian medical students [7,11]. The association of ergonomic aspects, such as the posture adopted by the computer user, with DES prevalence is poorly investigated [28], and the studies that consider this factor report inconclusive results regarding its effect on the development of eye problems [13,21,22].

Another DES-preventive factor that has been pointed out in this study is the observance—but not the awareness—of the 20-20-20 rule, which is similar to what studies carried out with Saudi Arabian university students found [7,13]. This finding confirms that to reduce DES prevalence, not only should awareness of this preventive measure be raised, but also, strategies should be adopted to encourage and monitor its implementation among university students. Although this study did not find breaks to be a DES-preventive factor, several studies that categorized this factor to determine whether participants took breaks or not agreed that taking breaks was a DES-preventive factor [18,24,25].

Finally, this study revealed that half of the people with DES wore glasses and that students who did not wear them had a lower risk of DES. Similarly, previous studies reported that the prevalence of this eye problem was higher among students with eye diseases and who wore glasses [8,9,12,29]. However, this study disagrees with other studies [7,19,24]. Something that could explain this discrepancy is that the young university population does not yet have a worn crystalline lens, and therefore, there is an absence of DES; however, this aspect needs to be addressed exhaustively in other research because it is an occupational disease that is not considered a priority but that, with the passage of time, can cause irreversible consequences for vision.

On the one hand, this study had some limitations. We classified participants as having DES based on the self-reporting of symptoms without medical evaluation. Additionally, since we did not exclude students with previous eye disorders, DES prevalence could have been overestimated. In addition, the study design only revealed association but not causality. Another limitation is related to the comparison of the results with those of international studies, which should be taken with caution because the study population varies depending on the setting. On the other hand, the strength of this study lies in the large number of students included. Despite the limitations, this paper provides the scientific community and educational institutions with important results on DES as an increasing health problem. Future research should consider using non-invasive test methods such as eye tracking to collect the eye movement data of the subjects and analyze the visual fatigue problem of online learning through eye-movement data.

## 5. Conclusions

In conclusion, this study provides evidence for high DES prevalence among nursing students. It also shows that the hours spent using electronic devices and screen brightness were risk factors of DES. However, sitting upright, wearing glasses, and following the 20-20-20 rule were preventive factors against DES.

To reduce this eye-health problem during virtual learning, universities should promote preventive strategies such as improving the ergonomics of the study environment, reducing exposure time, adjusting screen brightness to dull mode, and encouraging the practice of crucial eye-care measures such as the 20-20-20 rule.

## Figures and Tables

**Table 1 ijerph-20-05067-t001:** Comparison of conditions when using electronic devices in relation to the presence and absence of digital eye strain among nursing students.

	Digital Eye Strain		*p* Value *
	Absence	Presence	
	*n* (%)	*n* (%)	
Device most used			
Computer	76 (76.77)	551 (79.05)	0.716
Tablet	2 (2.02)	8 (1.15)	
Smartphone	21 (21.21)	138 (19.8)	
Daily hours of using device			
Between 1 and 4	27 (27.27)	98 (14.06)	0.001
More than 4	72 (72.73)	599 (85.94)	
Distance of the device from the eyes (cm)			
Less than 30	34 (34.34)	281 (40.32)	0.104
Between 30 and 50	56 (56.57)	385 (55.24)	
More than 50	9 (9.09)	31 (4.45)	
Posture when using the device			
Sitting bent over	44 (44.44)	470 (67.43)	<0.001
Sitting upright	52 (52.53)	210 (30.13)	
Laying down	3 (3.03)	17 (2.44)	
Screen brightness			
Dull	29 (29.29)	148 (21.23)	0.011
Bright	66 (66.67)	454 (65.14)	
Very bright	4 (4.04)	95 (13.63)	
Frequency of breaks			
Every 30 min	22 (22.22)	203 (29.12)	0.335
Every 1 h	17 (17.17)	105 (15.06)	
Every 2 h	28 (28.28)	152 (21.81)	
Every more than 2 h	32 (32.32)	237 (34.00)	
Duration of breaks (min)			
Between 1 and 5	26 (26.26)	214 (30.70)	0.585
Between 6 and 10	37 (37.37)	250 (35.87)	
Between 11 and 19	24 (24.24)	174 (24.96)	
More than 20	12 (12.12)	59 (8.46)	
Use of monitor filters			
No	68 (68.69)	525 (75.32)	0.058
Sometimes	14 (14.14)	107 (15.35)	
Yes	17 (17.17)	65 (9.33)	
Use of glasses			
Yes	36 (36.36)	349 (50.07)	0.014
No	63 (63.64)	348 (49.93)	
Awareness of the 20-20-20 rule			
Yes	20 (20.20)	85 (12.20)	0.041
No	79 (79.80)	612 (87.80)	
Observance of the 20-20-20 rule			
Yes	14 (14.14)	36 (5.16)	0.001
No	85 (85.86)	661 (4.84)	

* Chi-square test.

**Table 2 ijerph-20-05067-t002:** Logistic regression with variable selection for the associated factors of digital eye strain among nursing students.

	b	SE	*p*	OR [95% IC]
Intercept	1.17	0.49	0.02	3.23 [1.27–8.58]
Posture (sitting upright) (*)	−0.76	0.23	<0.01	0.47 [0.30–0.74]
Posture (laying down)	−0.59	0.68	0.39	0.56 [0.17–2.59]
Hours of usage (over 4 h) (†)	0.55	0.26	0.04	1.73 [1.02–2.86]
Observance of the 20-20-20 (no) (‡)	0.95	0.36	0.01	2.60 [1.25–5.20]
Brightness (bright) (§)	0.05	0.25	0.85	1.05 [0.63–1.71]
Brightness (very bright)	1.21	0.56	0.03	3.36 [1.23–11.8]
Glasses (no) (||)	−0.52	0.23	0.03	0.59 [0.37–0.93]
Filters (sometimes) (¶)	0.35	0.33	0.29	1.42 [0.76–2.81]
Filters (yes)	−0.64	0.32	0.05	0.53 [0.29–1.01]

b = unstandardized coefficient; SE = standard error; *p* = significance; OR = odds ratio. * The reference category for posture when using the devices was sitting bent over. † The reference category for the hours spent using devices was less than 4 h. ‡ The reference category for observance of the 20-20-20 rule was yes. § The reference category for screen brightness was dull. || The reference category for use of glasses was yes. ¶ The reference category for use of filters was no.

## Data Availability

The datasets generated or analyzed during this study are available from the corresponding author on reasonable request.

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
