# Peer review of "Digital Eye Strain among Peruvian Nursing Students: Prevalence and Associated Factors"

_ijerph, 2023, doi:10.3390/ijerph20065067_

Round 1
Reviewer 1 Report
This paper is a very valuable research. As we all know, since the COVID-19 pandemic, in order to avoid being infected by the virus, educational institutions require students to learn online. If they face computer screens or electronic devices for a long time, it will definitely bring about a series of vision problems. This paper obtains the experimental data through the self-evaluation of the subjects, and uses logical regression to analyze the relevant factors. As the author explained at the end, this study is only based on the self-evaluation of the subjects, which may lack objectivity to some extent. Future research can consider using non-invasive test methods such as eye tracking to collect the eye movement data of the subjects, and analyze the visual fatigue problem of the learners' online learning through the eye movement data.
Author Response
Thank you very much for your comment and suggestion. We have considered your suggestion appropriate and therefore it was incorporated at the end of the discussion.
Reviewer 2 Report
The article is solid. Suggestions for improvement.
1. The narrative on page 2, Lines 67-70 (starting with We followed...) are not necessary since IRB (Ethics) approval was granted. If the authors want to leave it in then they should explain how dignity, human rights, autonomy and responsibility was followed. Same for the next sentence with privacy, confidentiality, etc..
2. Please describe how the snowball sampling was done to reach 796. How was 796 identified as the sample size needed for the study? Or, is that the end of the snowball sampling
3. On page 3, Line 110 put a % after 19.47
Author Response
The article is solid. Suggestions for improvement.
Comment 1: The narrative on page 2, Lines 67-70 (starting with We followed...) are not necessary since IRB (Ethics) approval was granted. If the authors want to leave it in then they should explain how dignity, human rights, autonomy and responsibility was followed. Same for the next sentence with privacy, confidentiality, etc.
Response 1: We have removed the narrative mentioned.
Comment 2: Please describe how the snowball sampling was done to reach 796. How was 796 identified as the sample size needed for the study? Or, is that the end of the snowball sampling
Response 2: We have added the following statement on “Population, Sample and Selection criteria” in the materials and methods: “The estimated sample was 796 participants estimating a Power of at least 0.80, with a Type I Error probability set to be 0.05, and a small effect size. The sample and it was selected through by snowball sampling”
Comment 3: On page 3, Line 110 put a % after 19.47
Response 3: We have made the requested change. We also verified such mistake throughout the manuscript
Reviewer 3 Report
This is a very important topic and the paper is well written. However there are some issues that should be addressed before it can be published
1. In my opinion, some of the references are not the most convenient. Authors should try to find the original works for some of the citations. For example, they use the CVS_Q questionnaire but they did not cite the paper :
Seguí Mdel M, Cabrero-García J, Crespo A, Verdú J, Ronda E. A reliable and valid questionnaire was developed to measure computer vision syndrome at the workplace. J Clin Epidemiol. 2015 Jun;68(6):662-73. doi: 10.1016/j.jclinepi.2015.01.015. Epub 2015 Jan 28. PMID: 25744132.
3. The p values in table 2 should be better explained. They are related to what? For example, for "Daily hours of usage of the devices" the statistical significance refers to what? Is for the difference between the number of hours or for the presence/absence of DES?
4. Authors stated that "Something that could explain this discrepancy is 266 the absence of questions to learn whether the glasses were prescribed by an ophthalmologist or self-medicated. This aspect could entail a bias in determining the association of 268 this variable with DES.".
Is that possible to people self-medicated glasses? In my opinion, this is a forced explanation. Is that possible that the glasses are not well prescribed? Can they have other condition that was not diagnosed? This should be revised.
Author Response
This is a very important topic and the paper is well written. However there are some issues that should be addressed before it can be published
Comment 1: In my opinion, some of the references are not the most convenient. Authors should try to find the original works for some of the citations. For example, they use the CVS_Q questionnaire but they did not cite the paper :
Seguí Mdel M, Cabrero-García J, Crespo A, Verdú J, Ronda E. A reliable and valid questionnaire was developed to measure computer vision syndrome at the workplace. J Clin Epidemiol. 2015 Jun;68(6):662-73. doi: 10.1016/j.jclinepi.2015.01.015. Epub 2015 Jan 28. PMID: 25744132.
Response 1: We have made the requested change.
We have replaced the reference with the suggested alternative.
Comment 2: The p values in table 2 should be better explained. They are related to what? For example, for "Daily hours of usage of the devices" the statistical significance refers to what? Is for the difference between the number of hours or for the presence/absence of DES?
Response 2: We added the words suggested “daily” and “presence of DES” in order to clarify this result.
Comment 3: Authors stated that "Something that could explain this discrepancy is the absence of questions to learn whether the glasses were prescribed by an ophthalmologist or self-medicated. This aspect could entail a bias in determining the association of this variable with DES.".
Is that possible to people self-medicated glasses? In my opinion, this is a forced explanation. Is that possible that the glasses are not well prescribed? Can they have other condition that was not diagnosed? This should be revised
Response 3: We have revised this section of the discussion to say “Something that could explain this discrepancy is that the young university population does not yet have a worn crystalline lens and therefore there is an absence of ESD; however, this aspect needs to be addressed exhaustively in other research because it is an occupational disease that is not considered a priority and that with the passage of time can cause irreversible consequences in vision”.